# LOSS-AWARE BINARIZATION OF DEEP NETWORKS

**Lu Hou, Quanming Yao, James T. Kwok**
Department of Computer Science and Engineering
Hong Kong University of Science and Technology
Clear Water Bay, Hong Kong
{lhouab,qyaoaa,jamesk}@cse.ust.hk

## ABSTRACT

Deep neural network models, though very powerful and highly successful, are computationally expensive in terms of space and time. Recently, there have been a number of attempts on binarizing the network weights and activations. This greatly reduces the network size, and replaces the underlying multiplications to additions or even XNOR bit operations. However, existing binarization schemes are based on simple matrix approximations and ignore the effect of binarization on the loss. In this paper, we propose a proximal Newton algorithm with diagonal Hessian approximation that directly minimizes the loss w.r.t. the binarized weights. The underlying proximal step has an efficient closed-form solution, and the second-order information can be efficiently obtained from the second moments already computed by the Adam optimizer. Experiments on both feedforward and recurrent networks show that the proposed loss-aware binarization algorithm outperforms existing binarization schemes, and is also more robust for wide and deep networks.

## 1 INTRODUCTION

Recently, deep neural networks have achieved state-of-the-art performance in various tasks such as speech recognition, visual object recognition, and image classification (LeCun et al., 2015). Though powerful, the large number of network weights leads to space and time inefficiencies in both training and storage. For instance, the popular AlexNet, VGG-16 and Resnet-18 all require hundred of megabytes to store, and billions of high-precision operations on classification. This limits its use in embedded systems, smart phones and other portable devices that are now everywhere.

To alleviate this problem, a number of approaches have been recently proposed. One attempt first trains a neural network and then compresses it (Han et al., 2016; Kim et al., 2016). Instead of this two-step approach, it is more desirable to train and compress the network simultaneously. Example approaches include tensorizing (Novikov et al., 2015), parameter quantization (Gong et al., 2014), and binarization (Courbariaux et al., 2015; Hubara et al., 2016; Rastegari et al., 2016). In particular, binarization only requires one bit for each weight value. This can significantly reduce storage, and also eliminates most multiplications during the forward pass.

Courbariaux et al. (2015) pioneered neural network binarization with the BinaryConnect algorithm, which achieves state-of-the-art results on many classification tasks. Besides binarizing the weights, Hubara et al. (2016) further binarized the activations. Rastegari et al. (2016) also learned to scale the binarized weights, and obtained better results. Besides, they proposed the XNOR-network with both weights and activations binarized as in (Hubara et al., 2016). Instead of binarization, ternary-connect quantizes each weight to $\{-1, 0, 1\}$ (Lin et al., 2016). Similarly, the ternary weight network (Li & Liu, 2016) and DoReFa-net (Zhou et al., 2016) quantize weights to three levels or more. However, though using more bits allows more accurate weight approximations, specialized hardwares are needed for the underlying non-binary operations.

Besides the huge amount of computation and storage involved, deep networks are difficult to train because of the highly nonconvex objective and inhomogeneous curvature. To alleviate this problem, Hessian-free methods (Martens & Sutskever, 2012) use the second-order information by conjugate gradient. A related method is natural gradient descent (Pascanu & Bengio, 2014), which utilizes ge-

ometry of the underlying parameter manifold. Another approach uses element-wise adaptive learning rate, as in Adagrad (Duchi et al., 2011), Adadelta (Zeiler, 2012), RMSprop (Tieleman & Hinton, 2012), and Adam Kingma & Ba (2015). This can also be considered as preconditioning that rescales the gradient so that all dimensions have similar curvatures.

In this paper, instead of directly approximating the weights, we propose to consider the effect of binarization on the loss during binarization. We formulate this as an optimization problem using the proximal Newton algorithm (Lee et al., 2014) with a diagonal Hessian. The crux of proximal algorithms is the proximal step. We show that this step has a closed-form solution, whose form is similar to the use of element-wise adaptive learning rate. The proposed method also reduces to BinaryConnect (Courbariaux et al., 2015) and the Binary-Weight-Network (Hubara et al., 2016) when curvature information is dropped. Experiments on both feedforward and recurrent neural network models show that it outperforms existing binarization algorithms. In particular, BinaryConnect fails on deep recurrent networks because of the exploding gradient problem, while the proposed method still demonstrates robust performance.

**Notations:** For a vector $\mathbf{x}$, $\sqrt{\mathbf{x}}$ denotes the element-wise square root, $|\mathbf{x}|$ denotes the element-wise absolute value, $\|\mathbf{x}\|_p = (\sum_i |x_i|^p)^{\frac{1}{p}}$ is the $p$-norm of $\mathbf{x}$, $\mathbf{x} \succ 0$ denotes that all entries of $\mathbf{x}$ are positive, $\text{sign}(\mathbf{x})$ is the vector with $[\text{sign}(\mathbf{x})]_i = 1$ if $x_i \geq 0$ and $-1$ otherwise, and $\text{Diag}(\mathbf{x})$ returns a diagonal matrix with $\mathbf{x}$ on the diagonal. For two vectors $\mathbf{x}$ and $\mathbf{y}$, $\mathbf{x} \odot \mathbf{y}$ denotes the element-wise multiplication and $\mathbf{x} \oslash \mathbf{y}$ denotes the element-wise division. For a matrix $\mathbf{X}$, $\text{vec}(\mathbf{X})$ returns the vector obtained by stacking the columns of $\mathbf{X}$, and $\text{diag}(\mathbf{X})$ returns a diagonal matrix whose diagonal elements are extracted from diagonal of $\mathbf{X}$.

## 2 RELATED WORK

### 2.1 WEIGHT BINARIZATION IN DEEP NETWORKS

In a feedforward neural network with $L$ layers, let the weight matrix (or tensor in the case of a convolutional layer) at layer $l$ be $\mathbf{W}_l$. We combine the (full-precision) weights from all layers as $\mathbf{w} = [\mathbf{w}_1^\top, \mathbf{w}_2^\top, \ldots, \mathbf{w}_L^\top]^\top$, where $\mathbf{w}_l = \text{vec}(\mathbf{W}_l)$. Analogously, the binarized weights are denoted as $\hat{\mathbf{w}} = [\hat{\mathbf{w}}_1^\top, \hat{\mathbf{w}}_2^\top, \ldots, \hat{\mathbf{w}}_L^\top]^\top$. As it is essential to use full-precision weights during updates (Courbariaux et al., 2015), typically binarized weights are only used during the forward and backward propagations, but not on parameter update. At the $t$th iteration, the (full-precision) weight $\mathbf{w}_l^t$ is updated by using the backpropagated gradient $\nabla_l \ell(\hat{\mathbf{w}}^{t-1})$ (where $\ell$ is the loss and $\nabla_l \ell(\hat{\mathbf{w}}^{t-1})$ is the partial derivative of $\ell$ w.r.t. the weights of the $l$th layer). In the next forward propagation, it is then binarized as $\hat{\mathbf{w}}_l^t = \text{Binarize}(\mathbf{w}_l^t)$, where $\text{Binarize}(\cdot)$ is some binarization scheme.

The two most popular binarization schemes are BinaryConnect (Courbariaux et al., 2015) and Binary-Weight-Network (BWN) (Rastegari et al., 2016). In BinaryConnect, binarization is performed by transforming each element of $\mathbf{w}_l^t$ to $-1$ or $+1$ using the sign function:[1]

$$\text{Binarize}(\mathbf{w}_l^t) = \text{sign}(\mathbf{w}_l^t). \tag{1}$$

Besides the binarized weight matrix, a scaling parameter is also learned in BWN. In other words, $\text{Binarize}(\mathbf{w}_l^t) = \alpha_l^t \mathbf{b}_l^t$, where $\alpha_l^t > 0$ and $\mathbf{b}_l^t$ is binary. They are obtained by minimizing the difference between $\mathbf{w}_l^t$ and $\alpha_l^t \mathbf{b}_l^t$, and have a simple closed-form solution:

$$\alpha_l^t = \frac{\|\mathbf{w}_l^t\|_1}{n_l}, \quad \mathbf{b}_l^t = \text{sign}(\mathbf{w}_l^t), \tag{2}$$

where $n_l$ is the number of weights in layer $l$. Hubara et al. (2016) further binarized the activations as $\hat{\mathbf{x}}_l^t = \text{sign}(\mathbf{x}_l^t)$, where $\mathbf{x}_l^t$ is the activation of the $l$th layer at iteration $t$.

### 2.2 PROXIMAL NEWTON ALGORITHM

The proximal Newton algorithm (Lee et al., 2014) has been popularly used for solving composite optimization problems of the form

$$\min_{\mathbf{x}} f(\mathbf{x}) + g(\mathbf{x}),$$

---

[1] A stochastic binarization scheme is also proposed in (Courbariaux et al., 2015). However, it is much more computational expensive than (1) and so will not be considered here.

where $f$ is convex and smooth, and $g$ is convex but possibly nonsmooth. At iteration $t$, it generates the next iterate as

$$\mathbf{x}_{t+1} = \arg\min_{\mathbf{x}} \nabla f(\mathbf{x}_t)^\top (\mathbf{x} - \mathbf{x}_t) + (\mathbf{x} - \mathbf{x}_t)^\top \mathbf{H}(\mathbf{x} - \mathbf{x}_t) + g(\mathbf{x}),$$

where $\mathbf{H}$ is an approximate Hessian matrix of $f$ at $\mathbf{x}_t$. With the use of second-order information, the proximal Newton algorithm converges faster than the proximal gradient algorithm (Lee et al., 2014). Recently, by assuming that $f$ and $g$ have difference-of-convex decompositions (Yuille & Rangarajan, 2002), the proximal Newton algorithm is also extended to the case where $g$ is nonconvex (Rakotomamonjy et al., 2016).

## 3 LOSS-AWARE BINARIZATION

As can be seen, existing weight binarization methods (Courbariaux et al., 2015; Rastegari et al., 2016) simply find the closest binary approximation of $\mathbf{w}$, and ignore its effects to the loss. In this paper, we consider the loss directly during binarization. As in (Rastegari et al., 2016), we also binarize the weight $\mathbf{w}_l$ in each layer as $\hat{\mathbf{w}}_l = \alpha_l \mathbf{b}_l$, where $\alpha_l > 0$ and $\mathbf{b}_l$ is binary.

In the following, we make the following assumptions on $\ell$. (A1) $\ell$ is continuously differentiable with Lipschitz-continuous gradient, i.e., there exists $\beta > 0$ such that $\|\nabla\ell(\mathbf{u}) - \nabla\ell(\mathbf{v})\|_2 \le \beta \|\mathbf{u} - \mathbf{v}\|_2$ for any $\mathbf{u}, \mathbf{v}$; (A2) $\ell$ is bounded from below.

### 3.1 BINARIZATION USING PROXIMAL NEWTON ALGORITHM

We formulate weight binarization as the following optimization problem:

$$\min_{\hat{\mathbf{w}}} \quad \ell(\hat{\mathbf{w}}) \tag{3}$$

$$\text{s.t.} \quad \hat{\mathbf{w}}_l = \alpha_l \mathbf{b}_l, \ \alpha_l > 0, \ \mathbf{b}_l \in \{\pm 1\}^{n_l}, \ l = 1, \dots, L, \tag{4}$$

where $\ell$ is the loss. Let $C$ be the feasible region in (4), and define its indicator function: $I_C(\hat{\mathbf{w}}) = 0$ if $\hat{\mathbf{w}} \in C$, and $\infty$ otherwise. Problem (3) can then be rewritten as

$$\min_{\hat{\mathbf{w}}} \quad \ell(\hat{\mathbf{w}}) + I_C(\hat{\mathbf{w}}). \tag{5}$$

We solve (5) using the proximal Newton method (Section 2.2). At iteration $t$, the smooth term $\ell(\hat{\mathbf{w}}^t)$ is replaced by the second-order expansion

$$\ell(\hat{\mathbf{w}}^{t-1}) + \nabla\ell(\hat{\mathbf{w}}^{t-1})^\top(\hat{\mathbf{w}}^t - \hat{\mathbf{w}}^{t-1}) + \frac{1}{2}(\hat{\mathbf{w}}^t - \hat{\mathbf{w}}^{t-1})^\top \mathbf{H}^{t-1}(\hat{\mathbf{w}}^t - \hat{\mathbf{w}}^{t-1}),$$

where $\mathbf{H}^{t-1}$ is an estimate of the Hessian of $\ell$ at $\hat{\mathbf{w}}^{t-1}$. Note that using the Hessian to capture second-order information is essential for efficient neural network training, as $\ell$ is often flat in some directions but highly curved in others. By rescaling the gradient, the loss has similar curvatures along all directions. This is also called preconditioning in the literature (Dauphin et al., 2015a).

For neural networks, the exact Hessian is rarely positive semi-definite. This can be problematic as the nonconvex objective leads to indefinite quadratic optimization. Moreover, computing the exact Hessian is both time- and space-inefficient on large networks. To alleviate these problems, a popular approach is to approximate the Hessian by a diagonal positive definite matrix $\mathbf{D}$. One popular choice is the efficient Jacobi preconditioner. Though an efficient approximation of the Hessian under certain conditions, it is not competitive for indefinite matrices (Dauphin et al., 2015a). More recently, it is shown that equilibration provides a more robust preconditioner in the presence of saddle points (Dauphin et al., 2015a). This is also adopted by popular stochastic optimization algorithms such as RMSprop (Tieleman & Hinton, 2012) and Adam (Kingma & Ba, 2015). Specifically, the second moment $\mathbf{v}$ in these algorithms is an estimator of $\text{diag}(\mathbf{H}^2)$ (Dauphin et al., 2015b). Here, we use the square root of this $\mathbf{v}$, which is readily available in Adam, to construct $\mathbf{D} = \text{Diag}([\text{diag}(\mathbf{D}_1)^\top, \dots, \text{diag}(\mathbf{D}_L)^\top]^\top)$, where $\mathbf{D}_l$ is the approximate diagonal Hessian at layer $l$. In general, other estimators of $\text{diag}(\mathbf{H})$ can also be used.

At the $t$th iteration of the proximal Newton algorithm, the following subproblem is solved:

$$\min_{\hat{\mathbf{w}}^t} \quad \nabla\ell(\hat{\mathbf{w}}^{t-1})^\top(\hat{\mathbf{w}}^t - \hat{\mathbf{w}}^{t-1}) + \frac{1}{2}(\hat{\mathbf{w}}^t - \hat{\mathbf{w}}^{t-1})^\top \mathbf{D}^{t-1}(\hat{\mathbf{w}}^t - \hat{\mathbf{w}}^{t-1}) \tag{6}$$

$$\text{s.t.} \quad \hat{\mathbf{w}}_l^t = \alpha_l^t \mathbf{b}_l^t, \ \alpha_l^t > 0, \ \mathbf{b}_l^t \in \{\pm 1\}^{n_l}, \quad l = 1, \dots, L.$$

**Proposition 3.1** *Let* $\mathbf{d}_l^{t-1} \equiv diag(\mathbf{D}_l^{t-1})$*, and*

$$\mathbf{w}_l^t \equiv \hat{\mathbf{w}}_l^{t-1} - \nabla_l \ell(\hat{\mathbf{w}}^{t-1}) \oslash \mathbf{d}_l^{t-1}. \tag{7}$$

*The optimal solution of (6) can be obtained in closed-form as*

$$\alpha_l^t = \frac{\|\mathbf{d}_l^{t-1} \odot \mathbf{w}_l^t\|_1}{\|\mathbf{d}_l^{t-1}\|_1}, \quad \mathbf{b}_l^t = sign(\mathbf{w}_l^t). \tag{8}$$

**Theorem 3.1** *Assume that* $[\mathbf{d}_l^t]_k > \beta \ \forall l, k, t$*, the objective of* (5) *produced by the proximal Newton algorithm (with closed-form update of* $\hat{\mathbf{w}}^t$ *in Proposition 3.1) converges.*

Note that both the loss $\ell$ and indicator function $I_C(\cdot)$ in (5) are not convex. Hence, convergence analysis of the proximal Newton algorithm in (Lee et al., 2014), which is only for convex problems, cannot be applied. Recently, Rakotomamonjy et al. (2016) proposed a nonconvex proximal Newton extension. However, it assumes a difference-of-convex decomposition which does not hold here.

**Remark 3.1** *When* $\mathbf{D}_l^{t-1} = \lambda \mathbf{I}$*, i.e., the curvature is the same for all dimensions in the lth layer, (8) then reduces to the BWN solution in (2) In other words, BWN corresponds to using the proximal gradient algorithm, while the proposed method corresponds to the proximal Newton algorithm with diagonal Hessian. In composite optimization, it is known that the proximal Newton method is more efficient than the proximal gradient algorithm (Lee et al., 2014; Rakotomamonjy et al., 2016).*

**Remark 3.2** *When* $\alpha_l^t = 1$*, (8) reduces to* $sign(\mathbf{w}_l^t)$*, which is the BinaryConnect solution in (1).*

From (7) and (8), each iteration first performs gradient descent along $\nabla_l \ell(\hat{\mathbf{w}}^{t-1})$ with an adaptive learning rate $1 \oslash \mathbf{d}_l^{t-1}$, and then projects it to a binary solution. As discussed in (Courbariaux et al., 2015), it is important to keep a full-precision weight during training. Hence, we replace (7) by $\mathbf{w}_l^t \leftarrow \mathbf{w}_l^{t-1} - \nabla_l \ell(\hat{\mathbf{w}}^{t-1}) \oslash \mathbf{d}_l^{t-1}$. The whole procedure, which will be called Loss-Aware Binarization (LAB), is shown in Algorithm 1. In steps 5 and 6, following (Li & Liu, 2016), we first rescale input $\mathbf{x}_l^{t-1}$ to the $l$th layer with $\alpha_l$, so that multiplications in dot products and convolutions become additions.

While binarizing weights changes most multiplications to additions, binarizing both weights and activations saves even more computations as additions are further changed to XNOR bit operations (Hubara et al., 2016). Our Algorithm 1 can also be easily extended by binarizing the activations with the simple sign function.

## 3.2 Extension to Recurrent Neural Networks

The proposed method can be easily extended to recurrent neural networks. Let $\mathbf{x}_l$ and $\mathbf{h}_l$ be the input and hidden states, respectively, at time step (or depth) $l$. A typical recurrent neural network has a recurrence of the form $\mathbf{h}_l = \mathbf{W}_x \mathbf{x}_l + \mathbf{W}_h \sigma(\mathbf{h}_{l-1}) + \mathbf{b}$ (equivalent to the more widely known $\mathbf{h}_l = \sigma(\mathbf{W}_x \mathbf{x}_l + \mathbf{W}_h \mathbf{h}_{l-1} + \mathbf{b})$ (Pascanu et al., 2013) ). We binarize both the input-to-hidden weight $\mathbf{W}_x$ and hidden-to-hidden weight $\mathbf{W}_h$. Since weights are shared across time in a recurrent network, we only need to binarize $\mathbf{W}_x$ and $\mathbf{W}_h$ once in each forward propagation. Besides weights, one can also binarize the activations (of the inputs and hidden states) as in the previous section.

In deep networks, the backpropagated gradient takes the form of a product of Jacobian matrices (Pascanu et al., 2013). In a vanilla recurrent neural network,[2] for activations $\mathbf{h}_p$ and $\mathbf{h}_q$ at depths $p$ and $q$, respectively (where $p > q$), $\frac{\partial \mathbf{h}_p}{\partial \mathbf{h}_q} = \prod_{q < l \le p} \frac{\partial \mathbf{h}_l}{\partial \mathbf{h}_{l-1}} = \prod_{q < l \le p} \mathbf{W}_h^\top diag(\sigma'(\mathbf{h}_{l-1}))$. The necessary condition for exploding gradients is that the largest singular value $\lambda_1(\mathbf{W}_h)$ of $\mathbf{W}_h$ is larger than some given constant (Pascanu et al., 2013). The following Proposition shows that for any binary $\mathbf{W}_h$, its largest singular value is lower-bounded by the square root of its dimension.

**Proposition 3.2** *For any* $\mathbf{W} \in \{-1, +1\}^{m \times n}$ *(*$m \le n$*),* $\lambda_1(\mathbf{W}) \ge \sqrt{n}$*.*

---

[2]Here, we consider the vanilla recurrent neural network for simplicity. It can be shown that a similar behavior holds for the more commonly used LSTM.

---

**Algorithm 1** Loss-Aware Binarization (LAB) for training a feedforward neural network.

---

**Input:** Minibatch $\{(\mathbf{x}_0^t, \mathbf{y}^t)\}$, current full-precision weights $\{\mathbf{w}_l^t\}$, first moment $\{\mathbf{m}_l^{t-1}\}$, second moment $\{\mathbf{v}_l^{t-1}\}$, and learning rate $\eta^t$.

 1: **Forward Propagation**
 2: **for** $l = 1$ to $L$ **do**
 3: $\alpha_l^t = \frac{\|\mathbf{d}_l^{t-1} \odot \mathbf{w}_l^t\|_1}{\|\mathbf{d}_l^{t-1}\|_1}$;
 4: $\mathbf{b}_l^t = \mathrm{sign}(\mathbf{w}_l^t)$;
 5: rescale the layer-$l$ input: $\tilde{\mathbf{x}}_{l-1}^t = \alpha_l^t \mathbf{x}_{l-1}^t$;
 6: compute $\mathbf{z}_l^t$ with input $\tilde{\mathbf{x}}_{l-1}^t$ and binary weight $\mathbf{b}_l^t$;
 7: apply batch-normalization and nonlinear activation to $\mathbf{z}_l^t$ to obtain $\mathbf{x}_l^t$;
 8: **end for**
 9: compute the loss $\ell$ using $\mathbf{x}_L^t$ and $\mathbf{y}^t$;
10: **Backward Propagation**
11: initialize output layer's activation's gradient $\frac{\partial \ell}{\partial \mathbf{x}_L^t}$;
12: **for** $l = L$ to $2$ **do**
13: compute $\frac{\partial \ell}{\partial \mathbf{x}_{l-1}^t}$ using $\frac{\partial \ell}{\partial \mathbf{x}_l^t}$, $\alpha_l^t$ and $\mathbf{b}_l^t$;
14: **end for**
15: **Update parameters using Adam**
16: **for** $l = 1$ to $L$ **do**
17: compute gradients $\nabla_l \ell(\hat{\mathbf{w}}^t)$ using $\frac{\partial \ell}{\partial \mathbf{x}_l^t}$ and $\mathbf{x}_{l-1}^t$;
18: update first moment $\mathbf{m}_l^t = \beta_1 \mathbf{m}_l^{t-1} + (1 - \beta_1) \nabla_l \ell(\hat{\mathbf{w}}^t)$;
19: update second moment $\mathbf{v}_l^t = \beta_2 \mathbf{v}_l^{t-1} + (1 - \beta_2)(\nabla_l \ell(\hat{\mathbf{w}}^t) \odot \nabla_l \ell(\hat{\mathbf{w}}^t))$;
20: compute unbiased first moment $\hat{\mathbf{m}}_l^t = \mathbf{m}_l^t / (1 - \beta_1^t)$;
21: compute unbiased second moment $\hat{\mathbf{v}}_l^t = \mathbf{v}_l^t / (1 - \beta_2^t)$;
22: compute current curvature matrix $\mathbf{d}_l^t = \frac{1}{\eta^t} \left( \epsilon \mathbf{1} + \sqrt{\hat{\mathbf{v}}_l^t} \right)$;
23: update full-precision weights $\mathbf{w}_l^{t+1} = \mathbf{w}_l^t - \hat{\mathbf{m}}_l^t \oslash \mathbf{d}_l^t$;
24: update learning rate $\eta^{t+1} = \mathrm{UpdateRule}(\eta^t, t+1)$;
25: **end for**

---

Thus, with weight binarization as in BinaryConnect, the exploding gradient problem becomes more severe as the weight matrices are often large. On the other hand, recall that $\lambda_1(c\hat{\mathbf{W}}_h) = c\lambda_1(\hat{\mathbf{W}}_h)$ for any non-negative $c$. The proposed method alleviates this exploding gradient problem by adaptively learning the scaling parameter $\alpha_h$.

## 4 EXPERIMENTS

In this section, we perform experiments on the proposed binarization scheme with both feedforward networks (Sections 4.1 and 4.2) and recurrent neural networks (Sections 4.3 and 4.4).

### 4.1 FEEDFORWARD NEURAL NETWORKS

We compare the original full-precision network (without binarization) with the following weight-binarized networks: (i) BinaryConnect; (ii) Binary-Weight-Network (BWN); and (iii) the proposed Loss-Aware Binarized network (LAB). We also compare with networks having both weights and activations binarized:[3] (i) BinaryNeuralNetwork (BNN) (Hubara et al., 2016), the weight-and-activation binarized counterpart of BinaryConnect; (ii) XNOR-Network (XNOR) (Rastegari et al., 2016), the counterpart of BWN; (iii) LAB2, the counterpart of the proposed method, which binarizes weights using proximal Newton method and binarizes activations using a simple sign function.

The setup is similar to that in Courbariaux et al. (2015). We do not perform data augmentation or unsupervised pretraining. Experiments are performed on three commonly used data sets:

---

[3]We use the straight-through-estimator (Hubara et al., 2016) to compute the gradient involving the sign function.

1. *MNIST*: This contains $28 \times 28$ gray images from ten digit classes. We use $50000$ images for training, another $10000$ for validation, and the remaining $10000$ for testing. We use the 4-layer model:

$$784FC - 2048FC - 2048FC - 2048FC - 10SVM,$$

where $FC$ is a fully-connected layer, and $SVM$ is a L2-SVM output layer using the square hinge loss. Batch normalization, with a minibatch size 100, is used to accelerate learning. The maximum number of epochs is 50. The learning rate for the weight-binarized (resp. weight-and-activation-binarized) network starts at 0.01 (resp. 0.005), and decays by a factor of 0.1 at epochs 15 and 25.

2. *CIFAR-10*: This contains $32 \times 32$ color images from ten object classes. We use $45000$ images for training, another $5000$ for validation, and the remaining $10000$ for testing. The images are preprocessed with global contrast normalization and ZCA whitening. We use the VGG-like architecture:

$$(2 \times 128C3) - MP2 - (2 \times 256C3) - MP2 - (2 \times 512C3) - MP2 - (2 \times 1024FC) - 10SVM,$$

where $C3$ is a $3 \times 3$ ReLU convolution layer, and $MP2$ is a $2 \times 2$ max-pooling layer. Batch normalization, with a minibatch size of 50, is used. The maximum number of epochs is 200. The learning rate for the weight-binarized (resp. weight-and-activation-binarized) network starts at 0.03 (resp. 0.02), and decays by a factor of 0.5 after every 15 epochs.

3. *SVHN*: This contains $32 \times 32$ color images from ten digit classes. We use $598388$ images for training, another $6000$ for validation, and the remaining $26032$ for testing. The images are preprocessed with global and local contrast normalization. The model used is:

$$(2 \times 64C3) - MP2 - (2 \times 128C3) - MP2 - (2 \times 256C3) - MP2 - (2 \times 1024FC) - 10SVM.$$

Batch normalization, with a minibatch size of 50, is used. The maximum number of epochs is 50. The learning rate for the weight-binarized (resp. weight-and-activation-binarized) network starts at 0.001 (resp. 0.0005), and decays by a factor of 0.1 at epochs 15 and 25.

Since binarization is a form of regularization (Courbariaux et al., 2015), we do not use other regularization methods (like Dropout). All the weights are initialized as in (Glorot & Bengio, 2010). Adam (Kingma & Ba, 2015) is used as the optimization solver.

Table 1 shows the test classification error rates, and Figure 1 shows the convergence of LAB. As can be seen, the proposed LAB achieves the lowest error on *MNIST* and *SVHN*. It even outperforms the full-precision network on *MNIST*, as weight binarization serves as a regularizer. With the use of curvature information, LAB outperforms BinaryConnect and BWN. On *CIFAR-10*, LAB is slightly outperformed by BinaryConnect, but is still better than the full-precision network. Among the schemes that binarize both weights and activations, LAB2 also outperforms BNN and the XNOR-Network.

Table 1: Test error rates (%) for feedforward neural network models.

|  |  | *MNIST* | *CIFAR-10* | *SVHN* |
|---|---|---|---|---|
| (no binarization) | full-precision | 1.190 | 11.900 | 2.277 |
| (binarize weights) | BinaryConnect | 1.280 | **9.860** | 2.450 |
|  | BWN | 1.310 | 10.510 | 2.535 |
|  | LAB | **1.180** | 10.500 | **2.354** |
| (binarize weights and activations) | BNN | 1.470 | 12.870 | 3.500 |
|  | XNOR | 1.530 | 12.620 | 3.435 |
|  | LAB2 | **1.380** | **12.280** | **3.362** |

## 4.2 Varying the Number of Filters in CNN

As in Zhou et al. (2016), we study sensitivity to network width by varying the number of filters $\mathsf{K}$ on the *SVHN* data set. As in Section 4.1, we use the model

$$(2 \times \mathsf{K}C3) - MP2 - (2 \times 2\mathsf{K}C3) - MP2 - (2 \times 4\mathsf{K}C3) - MP2 - (2 \times 1024FC) - 10SVM.$$

Results are shown in Table 2. Again, the proposed LAB has the best performance. Moreover, as the number of filters increases, degradation due to binarization becomes less severe. This suggests

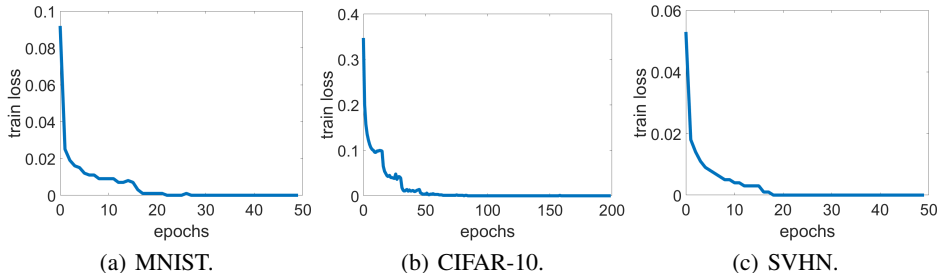

|               (a) MNIST.               |              (b) CIFAR-10.              |               (c) SVHN.               |

Figure 1: Convergence of LAB with feedforward neural networks.

that more powerful models (e.g., CNN with more filters, standard feedforward networks with more hidden units) are less susceptible to performance degradation due to binarization. We speculate that this is because large networks often have larger-than-needed capacities, and so are less affected by the limited expressiveness of binary weights. Another related reason is that binarization acts as regularization, and so contributes positively to the performance.

Table 2: Test error rates (%) on *SVHN*, for CNNs with different numbers of filters. Number in brackets is the difference between the errors of the binarized scheme and the full-precision network.

|               | K = 16        | K = 32        | K = 64        | K = 128       |
|---------------|---------------|---------------|---------------|---------------|
| full-precision | 2.738        | 2.585         | 2.277         | 2.146         |
| BinaryConnect | 3.200 (0.462) | 2.777 (0.192) | 2.450 (0.173) | 2.315 (0.169) |
| BWN           | 3.119 (0.461) | 2.743 (0.158) | 2.535 (0.258) | 2.319 (0.173) |
| LAB           | **3.050** (0.312) | **2.742** (0.157) | **2.354** (0.077) | **2.200** (0.054) |

## 4.3 RECURRENT NEURAL NETWORKS

In this section, we perform experiments on the popular long short-term memory (LSTM) (Hochreiter & Schmidhuber, 1997). Performance is evaluated in the context of character-level language modeling. The LSTM takes as input a sequence of characters, and predicts the next character at each time step. The training objective is the cross-entropy loss over all target sequences. Following Karpathy et al. (2016), we use two data sets (with the same training/validation/test set splitting): (i) Leo Tolstoy's *War and Peace*, which consists of 3258246 characters of almost entirely English text with minimal markup and has a vocabulary size of 87; and (ii) the source code of the *Linux Kernel*, which consists of 6206996 characters and has a vocabulary size of 101.

We use a one-layer LSTM with 512 cells. The maximum number of epochs is 200, and the number of time steps is 100. The initial learning rate is 0.002. After 10 epochs, it is decayed by a factor of 0.98 after each epoch. The weights are initialized uniformly in [0.08, 0.08]. After each iteration, the gradients are clipped to the range $[-5, 5]$, and all the updated weights are clipped to $[-1, 1]$. For the weight-and-activation-binarized networks, we do not binarize the inputs, as they are one-hot vectors in this language modeling task.

Table 3 shows the testing cross-entropy values. As in Section 4.1, the proposed LAB outperforms other weight binarization schemes, and is even better than the full-precision network on the *Linux Kernel* data set. BinaryConnect does not work well here because of the problem of exploding gradients (see Section 3.2 and more results in Section 4.4). On the other hand, BWN and the proposed LAB scale the binary weight matrix and perform better. LAB also performs better than BWN as curvature information is considered. Similarly, among schemes that binarize both weights and activations, the proposed LAB2 also outperforms BNN and XNOR-Network.

## 4.4 VARYING THE NUMBER OF TIME STEPS IN LSTM

In this experiment, we study the sensitivity of the binarization schemes with varying numbers of unrolled time steps ($TS$) in LSTM. Results are shown in Table 4. Again, the proposed LAB has the best performance. When $TS = 10$, the LSTM is relatively shallow, and all binarization schemes have similar performance as the full-precision network. When $TS \geq 50$, BinaryConnect fails,

Table 3: Testing cross-entropy values of LSTM.

| | | *War and Peace* | *Linux Kernel* |
|---|---|---|---|
| (no binarization) | full-precision | 1.268 | 1.329 |
| (binarize weights) | BinaryConnect | 2.942 | 3.532 |
| | BWN | 1.313 | 1.307 |
| | LAB | **1.291** | **1.305** |
| (binarize weights and activations) | BNN | 3.050 | 3.624 |
| | XNOR | 1.424 | 1.426 |
| | LAB2 | **1.376** | **1.409** |

while BWN and the proposed LAB perform better (as discussed in Section 3.2). Figure 2 shows the distributions of the hidden-to-hidden weight gradients for $TS = 10$ and 100. As can be seen, while all models have similar gradient distributions at $TS = 10$, the gradient values in BinaryConnect are much higher than those of the other algorithms for the deeper network ($TS = 100$).

Table 4: Testing cross-entropy on *War and Peace*, for LSTMs with different time steps ($TS$). Difference between cross-entropies of binarized scheme and full-precision network is shown in brackets.

| | $TS = 10$ | $TS = 50$ | $TS = 100$ | $TS = 150$ |
|---|---|---|---|---|
| full-precision | 1.527 | 1.310 | 1.268 | 1.249 |
| BinaryConnect | 1.528 (0.001) | 2.980 (1.670) | 2.942 (1.674) | 2.872 (1.623) |
| BWN | 1.532 (0.005) | 1.325 (0.015) | 1.313 (0.045) | 1.311 (0.062) |
| LAB | **1.527** (0.000) | **1.324** (0.014) | **1.291** (0.023) | **1.285** (0.036) |

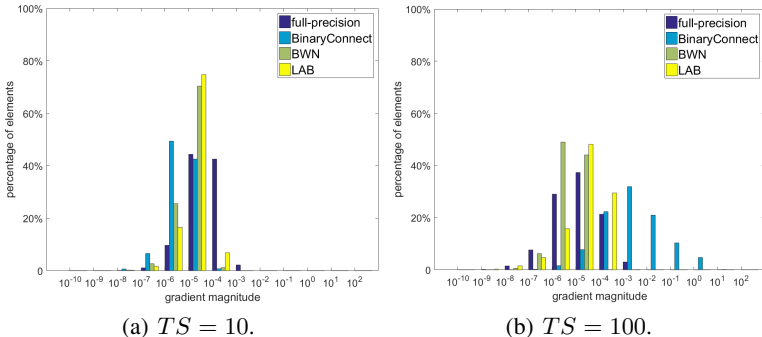

(a) $TS = 10$. (b) $TS = 100$.

Figure 2: Distribution of weight gradients on *War and Peace*, for LSTMs with different time steps.

Note from Table 4 that as the time step increases, all except BinaryConnect show better performance. However, degradation due to binarization also becomes more severe. This is because the weights are shared across time steps. Hence, error due to binarization also propagates across time.

## 5 CONCLUSION

In this paper, we propose a binarization algorithm that directly considers its effect on the loss during binarization. The binarized weights are obtained using proximal Newton algorithm with diagonal Hessian approximation. The proximal step has an efficient closed-form solution, and the second-order information in the Hessian can be readily obtained from the Adam optimizer. Experiments show that the proposed algorithm outperforms existing binarization schemes, has comparable performance as the original full-precision network, and is also robust for wide and deep networks.

### ACKNOWLEDGMENTS

This research was supported in part by the Research Grants Council of the Hong Kong Special Administrative Region (Grant 614513). We thank Yongqi Zhang for helping with the experiments, and developers of Theano (Theano Development Team, 2016), Pylearn2 (Goodfellow et al., 2013) and Lasagne. We also thank NVIDIA for the support of Titan X GPU.

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

## A   PROOF OF PROPOSITION 3.1

$$\nabla\ell(\hat{\mathbf{w}}^{t-1})^\top(\hat{\mathbf{w}}^t - \hat{\mathbf{w}}^{t-1}) + \frac{1}{2}(\hat{\mathbf{w}}^t - \hat{\mathbf{w}}^{t-1})^\top \mathbf{D}^{t-1}(\hat{\mathbf{w}}^t - \hat{\mathbf{w}}^{t-1})$$

$$= \frac{1}{2}\sum_{l=1}^{L}\left(\sqrt{(\mathbf{d}_l^{t-1})^\top}\left(\hat{\mathbf{w}}_l^t - (\hat{\mathbf{w}}_l^{t-1} - \nabla_l\ell(\hat{\mathbf{w}}^{t-1})\oslash\mathbf{d}_l^{t-1})\right)\right)^2 + c_1$$

$$= \frac{1}{2}\sum_{l=1}^{L}\left(\sqrt{(\mathbf{d}_l^{t-1})^\top}(\hat{\mathbf{w}}_l^t - \mathbf{w}_l^t)\right)^2 + c_1$$

$$= \frac{1}{2}\sum_{l=1}^{L}\left(\sqrt{(\mathbf{d}_l^{t-1})^\top}(\alpha_l^t\mathbf{b}_l^t - \mathbf{w}_l^t)\right)^2 + c_1,$$

where $c_1 = -\frac{1}{2}\left(\sqrt{(\mathbf{d}_l^{t-1})^\top}(\nabla_l\ell(\hat{\mathbf{w}}^{t-1})\oslash\mathbf{d}_l^{t-1})\right)^2$. Since $\alpha_l^t > 0, \mathbf{d}_l^t \succ \mathbf{0}, \forall l = 1, 2, \dots, L$, we have $\mathbf{b}_l^t = \mathrm{sign}(\mathbf{w}_l^t)$. Moreover,

$$\frac{1}{2}\sum_{l=1}^{L}(\sqrt{(\mathbf{d}_l^{t-1})^\top}(\alpha_l^t\mathbf{b}_l^t - \mathbf{w}_l^t))^2 + c_1 = \frac{1}{2}\sum_{l=1}^{L}\left(\sqrt{(\mathbf{d}_l^{t-1})^\top}(|\alpha_l^t\mathbf{1} - |\mathbf{w}_l^t||)\right)^2 + c_1$$

$$= \sum_{l=1}^{L}\frac{1}{2}\|\mathbf{d}_l^{t-1}\|_1(\alpha_l^t)^2 - \|\mathbf{d}_l^{t-1}\odot\mathbf{w}_l^t\|_1\alpha_l^t + c_2,$$

where $c_2 = c_1 - \frac{1}{2}\frac{\|\mathbf{d}_l^{t-1}\odot\mathbf{w}_l^t\|_1^2}{\|\mathbf{d}_l^{t-1}\|_1}$. Thus, the optimal $\alpha_l^t$ is $\frac{\|\mathbf{d}_l^{t-1}\odot\mathbf{w}_l^t\|_1}{\|\mathbf{d}_l^{t-1}\|_1}$.

## B   PROOF OF THEOREM 3.1

Let $\boldsymbol{\alpha} = [\alpha_1^t \dots, \alpha_L^t]^\top$, and denote the objective in (3) by $F(\hat{\mathbf{w}}, \boldsymbol{\alpha})$. As $\hat{\mathbf{w}}^t$ is the minimizer in (6), we have

$$\ell(\hat{\mathbf{w}}^{t-1}) + \nabla\ell(\hat{\mathbf{w}}^{t-1})^\top(\hat{\mathbf{w}}^t - \hat{\mathbf{w}}^{t-1}) + \frac{1}{2}(\hat{\mathbf{w}}^t - \hat{\mathbf{w}}^{t-1})^\top\mathbf{D}^{t-1}(\hat{\mathbf{w}}^t - \hat{\mathbf{w}}^{t-1}) \le \ell(\hat{\mathbf{w}}^{t-1}). \quad (9)$$

From Assumption A1, we have

$$\ell(\hat{\mathbf{w}}^t) \le \ell(\hat{\mathbf{w}}^{t-1}) + \nabla\ell(\hat{\mathbf{w}}^{t-1})^\top(\hat{\mathbf{w}}^t - \hat{\mathbf{w}}^{t-1}) + \frac{\beta}{2}\left\|\hat{\mathbf{w}}^t - \hat{\mathbf{w}}^{t-1}\right\|_2^2. \quad (10)$$

Using (9) and (10), we obtain

$$\ell(\hat{\mathbf{w}}^t) \le \ell(\hat{\mathbf{w}}^{t-1}) - \frac{1}{2}(\hat{\mathbf{w}}^t - \hat{\mathbf{w}}^{t-1})^\top(\mathbf{D}^{t-1} - \beta\boldsymbol{I})(\hat{\mathbf{w}}^t - \hat{\mathbf{w}}^{t-1})$$

$$\le \ell(\hat{\mathbf{w}}^{t-1}) - \frac{\min_{k,l}([d_l^{t-1}]_k - \beta)}{2}\left\|\hat{\mathbf{w}}^t - \hat{\mathbf{w}}^{t-1}\right\|_2^2.$$

Let $c_3 = \min_{k,l,t}([d_l^{t-1}]_k - \beta) > 0$. Then,

$$\ell(\hat{\mathbf{w}}^t) \le \ell(\hat{\mathbf{w}}^{t-1}) - \frac{c_3}{2}\left\|\hat{\mathbf{w}}^t - \hat{\mathbf{w}}^{t-1}\right\|_2^2. \quad (11)$$

From Assumption A2, $\ell$ is bounded from below. Together with the fact that $\{\ell(\hat{\mathbf{w}}^t)\}$ is monotonically decreasing from (11), the sequence $\{\ell(\hat{\mathbf{w}}^t)\}$ converges, thus the sequence $\{F(\hat{\mathbf{w}}^t, \boldsymbol{\alpha}^t)\}$ also converges.

## C   PROOF OF PROPOSITION 3.2

Let the singulars values of $\mathbf{W}$ be $\lambda_1(\mathbf{W}) \ge \lambda_2(\mathbf{W}) \ge \cdots \ge \lambda_m(\mathbf{W})$.

$$\lambda_1^2(\mathbf{W}) \ge \frac{1}{m}\sum_{i=1}^{m}\lambda_i^2(\mathbf{W}) = \frac{1}{m}\|\mathbf{W}\|_F^2 = \frac{1}{m}mn = n.$$

Thus, $\lambda_1(\mathbf{W}) \ge \sqrt{n}$.

