# Peer review of "Loss-aware Binarization of Deep Networks"

_ICLR 2017 — accepted_

[Official Review · AnonReviewer2 · rating 7 · confidence 3 · 14 Dec 2016]
**Inclusion of loss in the binarization of deep networks using a proximal Newton algorithm, where the proximal term is an indicator function for the binarized weights. This is the reasonable next step on the recently published works for training and compressing the network simultaneously (Binary Connect and Binary weight Network). Shown that it outperforms those binarization schemes on Mnist, svhn and RNN LM task**

Taking into account the loss in the binarization step through a proximal Newton algorithm is a nice idea. This is at least one approach to bringing in the missing loss in the binarization step, which has recently gone from a two step process of train and binarize to a single step simultaneous train/compress. Performance on a few small tasks show the benefit. It would be nice to see some results on substantial networks and tasks which really need compression on embedded systems (a point made in the introduction). Is it necessary to discuss exploding/vanishing gradients when the RNN experiments are carried out by an LSTM, and handled by the cell error carousel? We see the desire to tie into proposition 2, but not clear that the degradation we see in the binary connect is related. Adam is used in the LSTM optimization, was gradient clipping really needed, or is the degradation of binary connect simply related to capacity? For proposition 3.1, theorem 3.1 and proposition 3.2 put the pointers to proofs in appendix.

[Public Comment · (anonymous) · 20 Dec 2016]
**Is BWN also loss-aware?**

It seems not so fair to claim that the paper is the first "loss-aware" approach. The same optimization objective as BWN is used and the only difference is that the paper uses proximal Newton method instead of proximal gradient descent. It will be more convincing if the author could compare more with BWN. (In fact, BWN also has good performance in RNN)

[Official Review · AnonReviewer3 · rating 7 · confidence 4 · 20 Dec 2016]
**Novel 2nd order loss-aware binarization method for neural networks. Optimization performance is evaluated through the test error proxy.**

The paper presents a second-order method for training a neural networks while ensuring at the same time that weights (and activations) are binary. Through binarization, the method aims to achieve model compression for subsequent deployment on low-memory systems. The method is abbreviated BPN for "binarization using proximal Newton algorithm".

The method incorporates the supervised loss function directly in the binarization procedure, which is an important and desirable property. (Authors mention that existing weight binarization methods ignore the effect of binarization to the loss.) The method is clearly described and related analytically to the previously proposed weight binarization methods.

The experiments are extensive with multiple datasets and architectures, and demonstrate the generally higher performance of the proposed approach.

A minor issue with the feed-forward network experiments is that only test errors are reported. Such information does not really give evidence for the higher optimization performance. (see also comment "RE: AnonReviewer3's questions" stating that all baselines achieve near perfect training accuracy.) Making the optimization problem harder (e.g. by including an explicit regularizer into the training objective, or by using a data extension scheme), and monitoring the training objective instead of the test error could be a more direct way of demonstrating superior optimization performance.

The superiority of BPN is however becoming more clearly apparent in the subsequent LSTM experiments.

[Official Review · AnonReviewer1 · rating 7 · confidence 3 · 27 Dec 2016 (modified: 21 Jan 2017)]
**No Title**

This paper proposed a proximal (quasi-) Newton’s method to learn binary DNN. The main contribution is to combine pre-conditioning with binarization in a proximal framework. It is interesting to have a proximal Newton’s method to interpret the different DNN binarization schemes. This gives a new interpretation of existing approaches. However, the theoretical analysis is not very convincing or useful. The formulated optimization problem (3)-(4) is essentially a mixed integer programming. Even though the paper treats the integer part as a constraint and address it in proximal operators, the constraint set is still discrete and there is no guarantee that the proximal Newton algorithm could converge under practically useful conditions. In practice it is hard to verify the assumption [d_t^t]_k > \beta in Theorem 3.1. This relation could be hard to hold in DNN as the loss surface could be extremely complicated.

[Final Decision · Program Chairs · 06 Feb 2017]
**ICLR committee final decision**

It's a simple contribution supported by empirical and theoretical analyses. After some discussion, all reviewers viewed the paper favourably.